# Early-Life Supplementation of Bovine Milk Osteopontin Supports Neurodevelopment and Influences Exploratory Behavior

**DOI:** 10.3390/nu12082206

**Published:** 2020-07-24

**Authors:** Sangyun Joung, Joanne E. Fil, Anne B. Heckmann, Anne S. Kvistgaard, Ryan N. Dilger

**Affiliations:** 1University of Illinois, Neuroscience Program, Urbana, IL 61801, USA; sjoung2@illinois.edu (S.J.); jfil2@illinois.edu (J.E.F.); 2Arla Foods Ingredients, Arla Foods Ingredients Group P/S, DK-8260 Viby, Denmark; anhec@arlafoods.com (A.B.H.); askv@arlafoods.com (A.S.K.); 3Department of Animal Sciences, University of Illinois, Urbana, IL 61801, USA; 4Division of Nutritional Sciences, University of Illinois, Urbana, IL 61801, USA

**Keywords:** osteopontin, breastmilk, pig, brain, neurodevelopment, magnetic resonance imaging, novel object recognition, pediatric nutrition

## Abstract

Introduction: Osteopontin (OPN) is a whey protein found at high concentration in human milk and is involved in processes such as bone cell proliferation and differentiation. Milk OPN has shown to be involved in various aspects of development, including the immune system and gut health. However, the influence of dietary bovine milk OPN inclusion on brain and cognitive development has not been studied extensively until recently. This research examines whether dietary supplementation of bovine milk OPN supports brain and cognitive development in the translational pig model. Methods: From postnatal day (PND) 2 to 34, twenty-one intact male pigs were provided ad libitum access to one of two dietary treatments, a standard soy protein isolate-based milk replacer to serve as a control diet (*n* = 11) and the same base diet supplemented with bovine milk OPN to serve as a test diet (*n* = 10). In addition to growth and health outcomes, recognition memory was tested using the novel object recognition (NOR) task from PND 28 to 32, and magnetic resonance imaging was conducted at PND 34 to evaluate brain development. Results: No dietary effects were observed for growth performance or health indices. For the behavioral analysis, pigs that received the test diet exhibited shorter (*p* < 0.05) latency to the first object visited compared with pigs fed the control diet. Although the control group exhibited novelty preference, there was no difference in recognition index between dietary groups. Neuroimaging outcomes revealed increased (*p* < 0.05) relative brain volumes of the corpus callosum, lateral ventricle, left and right internal capsule, left and right putamen-globus pallidus, and right hippocampus, and right cortex in the test group. Diffusion tensor imaging revealed higher (*p* < 0.05) radial diffusivity in the corpus callosum and lower (*p* < 0.05) fractional anisotropy in pigs provided the test diet. Conclusion: Dietary supplementation of bovine milk OPN increased the relative volume of several brain regions and altered behaviors in the NOR task. Underlying mechanisms of bovine milk OPN influencing the development of brain structures and additional behaviors warrant further investigation.

## 1. Introduction

Human milk is well-known to be the optimal source of nutrition and supports neonatal health and development. Human milk contains not only essential macronutrients but also bioactive compounds such as proteins, lipids, and oligosaccharides [1,2]. Despite continuous improvements of infant formulas to mimic the composition of human milk, growth, health, and neurodevelopment outcomes of formula-fed infants are different to those of breast-fed infants [3]. Previous research has shown that the difference in bioactive compounds, such as proteins, in human milk and infant formula may explain the difference between breast fed infants and formula-fed infants in growth, health outcomes, microbiome, and immunological development [4,5,6]. Human milk contains a relatively high concentration of osteopontin (OPN) compared with commercial infant formulas [7]. As a highly phosphorylated glycoprotein with an integrin binding sequence, OPN is synthesized in various tissues and exists in many different isoforms due to post-translational modulation and splicing of the protein. Therefore, OPN is a multifunctional protein involved in processes such as bone cell proliferation and differentiation, bone remodeling, and anti-inflammatory responses [8,9].

Milk OPN is involved in diverse bioactivities during early life [10]. It can increase resistance against various pathogens by binding to integrins and consequently preventing binding of virus motif to integrins in target cells [11]. Breastfeeding is known to stimulate T helper cells, and OPN has also been shown to play an important role in inducing T helper cell type immunity [12,13]. As bovine milk OPN has become commercially available to supplement into infant formula, effects of dietary OPN inclusion on overall health outcomes and immune development have been compared to those of breast-fed infants [6]. Infants receiving OPN-supplemented formula experienced lower rate of fever incidences compared to infants on regular formula. Interestingly, the fever incidence rate was more similar to that of breast-fed infants. At four months of age, the regular formula-fed infants exhibited a higher concentration of tumor necrosis factor-α, a pro-inflammatory cytokine, than either infants receiving OPN-supplemented formula, who were shown to have concentrations indistinguishable from breast-fed infants [6].

In addition, a large portion of bovine milk OPN reaches the small intestine intact because the protein can withstand the digestive process in neonates relatively well [14]. Once the bovine milk OPN molecules reach the small intestine, they are thought to have bioactive functions, which include improving intestinal health and development. The ability for the oral administration of milk OPN to improve intestinal development, compared with both human milk and regular infant formula, has been studied in infant rhesus monkeys [15]. Infant rhesus monkeys who were fed infant formula supplemented with bovine milk OPN did not exhibit changes in circulating immune cell profiles or overall health and growth performance, but the addition of bovine milk OPN altered gene expression in the intestine [15]. A number of genes expressed in the intestine were different between breast-fed and formula-fed groups of rhesus infants. Addition of OPN to the formula noticeably narrowed the gap, suggesting oral OPN administration shifted intestinal gene expression patterns of formula-fed monkeys to make them more similar to the breast-fed group [8]. These findings collectively support the beneficial influence of bovine milk OPN in immune development.

Breasting-feeding has been shown to have beneficial effects not only on overall growth performance and gut health, but also on brain growth and health [16,17], and the role of milk OPN on brain development has been a topic of interest in recent literature. Milk OPN has been found to pass the blood–brain barrier in mouse pups, where it significantly increased the expression of myelination-related proteins such as myelin basic protein and myelin-associated glycoprotein [18]. Whereas dietary milk-derived OPN has not been studied in this context, endogenous OPN synthesis has been shown to play an important role in re-myelination for neurodegenerative diseases such as Alzheimer’s disease [19], and brain astrocytes have been shown to upregulate expression of OPN during re-myelination processes in mice, suggesting it may be involved in a repair-related role [20]. Expression of OPN in the cerebrospinal fluid and brain has been shown to positively correlate with cognitive decline in Alzheimer’s disease patients as well, protecting against the progression of neurodegeneration [19,21,22].

Although various studies have been performed to examine the effects of dietary OPN on immune functions, gut health and myelination during early life, the influence of bovine milk OPN on neonatal brain and cognitive development, especially during infancy, has not been extensively studied until recently [18,23]. Thus, the objective of the present study was to assess the effects of bovine milk OPN supplementation on neonatal brain and cognitive development using the translational pig model. The young pig is a powerful biomedical model for studying early-life nutritional interventions because pigs have strikingly similar nutrient requirements to humans during infancy, growth, and reproduction [24]. Additionally, humans share far more immune-system related genes and proteins with pigs than with mice [25]. This is likely due to their comparable enteric microbiota [26] and gastrointestinal physiology [27] with humans. Moreover, the morphology and peak brain growth of the pig [28,29] most closely resembles that of humans rather than rodents. Endogenous OPN has been identified in porcine cells, and it is 40% identical to OPN sequences for the rat, mouse, human, and cow [30]. Moreover, porcine OPN expression has been suggested to be present at the maternal–fetal interface, playing a crucial role in embryo implantation [31], and OPN expression in pigs is also known to be upregulated when exposed to infectious agents [32]. Additionally, the addition of OPN-enriched ingredients to milk replacer elicited beneficial effects on immune and gut responses during an inflammatory challenge in preterm pigs [33]. Collectively, these previous studies on OPN in pigs suggest that using the pig as a preclinical model is appropriate when testing the effects of dietary OPN ingestion. We aimed to evaluate the effects of bovine milk OPN supplementation on a variety of developmental outcomes, including microbial metabolite for evidence of a possible gut-brain-axis mechanism, neuroimaging for quantifying brain structure, and the novel object recognition (NOR) task for assessing cognitive behaviors. We hypothesized that pigs receiving bovine milk OPN would have more myelination occurring in the brain and exhibit improved performance on cognitive tasks compared with pigs receiving a diet devoid of OPN.

## 2. Materials and Methods

### 2.1. Animals and Housing

All animals and experimental procedures were conducted in accordance with the National Research Council Guide for the Care and Use of Laboratory Animals and approved by the University of Illinois at Urbana-Champaign Institutional Animal Care and Use Committee. Approval for this research project was confirmed on 6 February 2018 under the title Nutrition and Brain Development in Young Pigs. Twenty-eight intact (i.e., not castrated) male pigs were obtained from a commercial swine farm on postnatal day (PND) 2 and transferred to the University of Illinois Piglet Nutrition and Cognition Laboratory where they were artificially reared until PND 33 or 34. The trial was completed using 2 cohorts of pigs that were offset with respect to the study enrollment date by exactly one week, with a total of 14 replicate pigs per diet group. Pigs were selected from 4 separate litters within each cohort and allotted to each diet group based on litter of origin and initial body weight. All pigs were provided a single dose of prophylactic antibiotic (Excede Zoetis, Parsippany, NJ, USA) administered at 5.0 mg/kg body weight on the day of birth and two doses of 5 mL *Clostridium perfringens* antitoxin C and D (Colorado Serum Company, Denver, CO, USA; one dose given subcutaneously and the other given orally) upon arrival at the research facility at PND 2. Per veterinary recommendation, all pigs were treated with a single dose of 0.5 mL Penicillin orally at PND 6, and 2 doses of 1 mL oral antibiotic (SpectoGard, Bimenda Inc., LeSueur, MN, USA) for 3 consecutive days starting at PND 7.

Pigs were individually housed in custom rearing units, as previously described [34], which allowed the pigs to see, hear, and smell neighboring pigs but kept them separate to allow for discrete delivery of dietary treatments and quantitative measurement of milk replacer intake. Lights were automatically turned on and off at 08:00 h and 20:00 h, respectively. Twice daily health checks and daily pig body weights were recorded to track the presence of diarrhea, lethargy, weight loss, or vomiting as clinical indicators.

### 2.2. Dietary Groups and Feeding Procedures

A control milk replacer formula was designed to meet all nutritional requirements of the young pig [35] and was based on soy protein isolate in lieu of milk-based ingredients that would inherently contain OPN. The test milk replacer formula was based on the control formula but replaced a small portion of the soy protein isolate with a whey-based, bovine-derived OPN product (Lacprodan^®^ OPN-10; Arla Foods Ingredients; Viby, Denmark). The OPN-supplemented formula was designed to provide 250 mg of OPN/L of reconstituted milk replacer, and this concentration of OPN was chosen to mimic human milk that contains approximately 100–300 mg of OPN/L [7].

All individuals involved in conducting and analyzing the experimental results remained blinded to the identity of the dietary treatments until final analyses had been completed. Milk replacers were reconstituted fresh daily at 200 g of dry powder per 800 g of tap water. Pigs were fed ad libitum using an automated milk replacer delivery system from 10:00 to 06:00 h the next day [36]. Leftover milk from the previous day and individual pig body weights were recorded daily. The remaining volume of milk was subtracted from the initial volume provided to quantify milk disappearance over the 20 h feeding period, which will henceforth be referred to as the milk intake.

### 2.3. Behavioral Testing

Novel object recognition (NOR), described in detail previously [37], was used to assess recognition memory. Testing consisted of a habituation phase, a sample phase, and a test phase. During the habituation phase (PND28,29, each pig was placed in an empty testing arena for 10 min each day for two days leading up to the sample phase. In the sample phase (PND 30), the pig was placed in the arena containing two identical objects and given 5 min for exploration. After a delay of 48 h, the pig was returned to the arena for the test phase. During the test phase (PND 32), the pig was placed in the arena containing one object from the sample phase as well as a novel object and allowed to explore for 5 min. Between trials, objects were removed, immersed in hot water with detergent, and rubbed with a towel to mitigate odor, and the arena was sprayed with water to remove urine and feces. Objects chosen had a range of characteristics (i.e., color, texture, shape, and size); however, the novel and sample objects only differed in shape and size. Only objects previously shown to elicit a null preference were used for testing. All pigs that completed the study were tested on the NOR task. The recognition index, the proportion of time spent with the novel object compared to total exploration of both objects, was compared to a chance performance value of 0.50 to assess recognition memory. Values greater than 0.50 are indicative of a novelty preference, and thus recognition memory.

### 2.4. Magnetic Resonance Imaging

All piglets underwent magnetic resonance imaging (MRI) procedures on PND 33 or 34 at the Beckman Institute Biomedical Imaging Center using a Siemens MAGNETOM Prisma 3T MRI, with a custom made 8-channel piglet head coil. The pig neuroimaging protocol included a magnetization-prepared rapid gradient-echo (MPRAGE) sequence and diffusion tensor imaging (DTI) to assess brain macrostructure and microstructure, respectively. In preparation for MRI procedures, anesthesia was induced using an intramuscular injection of a telazol:ketamine:xylazine mixed solution (50.0 mg tiletamine plus 50.0 mg of zolazepam reconstituted with 2.50 mL ketamine (100 g/L) and 2.50 mL xylazine (100 g/L); Fort Dodge Animal Health, Overland Park, KS, USA) at 0.03 mL/kg of body weight. Once completely anesthetized, pigs were placed in a supine position in the MRI machine. Anesthesia was maintained by inhalation of 2% isoflurane/98% oxygen throughout the entire procedure. Oxygen saturation levels and heart rate were monitored using a pulse oximeter with an infrared sensor that was clipped on the pig’s tail or left hind hoof. Observational records of heart rate, partial pressure of oxygen, and percent of isoflurane were recorded every 5 min after anesthetic induction. Total scan time for each pig was approximately 60 min. Details of individual imaging sequences are briefly described below.

#### 2.4.1. Structural MRI Acquisition and Analysis

A T1-weighted MPRAGE sequence was used to obtain anatomic images of the piglet brain, with a 0.6 mm isotropic voxel size. The following sequence specific parameters were used to acquire T1-weighted MPRAGE data: TR = 2000 ms; TE = 2.05 ms; 288 slices; flip angle = 9°. All data generated used a population-averaged pig brain atlas [38] and brains were manually extracted as previously described [39]. After brain extraction, images were co-registered to the Pig MRI Atlas using the “Coregister: Estimate” toolbox in SPM12 (University College London, London, UK). Next, images were linearly registered to the atlas utilizing FLIRT from FMRIB Software Library (FSL) and then nonlinearly registered to the atlas utilizing FNIRT. An inverse of the output warp from FNIRT was applied on the atlas regions of interest (ROI) to bring the ROI from atlas space into individual subject space. All ROI were expressed in absolute volumes as well as percent of total brain volume (% TBV), to account for absolute whole-brain volume. The following equation was used: (region of interest absolute volume)/(total brain absolute volume) × 100, within subject.

#### 2.4.2. Diffusion Tensor Imaging Acquisition and Analysis

Diffusion tensor imaging (DTI) was used to assess white matter maturation and axonal tract integrity using b value = 1000 s/mm^2^ and 2000 s/mm^2^ across 30 directions and a 1.6 mm isotropic voxel. Diffusion-weighted echo planar images were assessed in FSL for fractional anisotropy (FA), mean diffusivity (MD), axial diffusivity (AD), and radial diffusivity (RD) using methods previously described [39]. Assessment was performed over the following regions of interest: caudate, corpus callosum, cerebellum, left and right hippocampi, left and right internal capsule, left and right sides of the brain, thalamus, and DTI generated white matter, and atlas generated white matter was performed using a customized pig analysis pipeline and the FSL software package. Atlas-generated white matter indicates the use of white matter prior probability maps from the pig brain atlas that were used as a region of interest mask. Likewise, DTI-generated white matter indicates a threshold of 0.2 was applied to FA values, thus restricting analysis to white matter tracts. A threshold of 0.45 was applied to each mask of the ROI, and the data and a FA threshold of 0.5 was applied to ensure that we only measured white matter in that region of interest.

#### 2.4.3. Voxel-Based Morphometry

Voxel-based morphometry (VBM) is a technique used to compare the distribution of gray matter and white matter between different treatments. Clusters of voxels indicate if a certain group has differential concentrations of gray matter or white matter in a certain location. Assessment of gray matter and white matter tissue concentrations was performed using SPM12 software (Wellcome Department of Clinical Neurology, London, UK). First, the Diffeomorphic Anatomical Registration using Exponentiated Lie Algebra (DARTEL) toolbox was used to create proper flow fields and warped files necessary for VBM. Pig-specific modification to the toolbox included changing the bounding box of −30.1 to 30.1, −35 to 44.8, −28 to 31.5; and a voxel size of 0.6 mm^3^. Warped files were then applied to each pig’s gray and white matter. The modulated data were smoothed with a 4 mm full-width half maximum, then subjected to the statistical non-parametric (SnPM13) methods toolbox for proper analysis.

### 2.5. Sample Collection and Assessment

#### 2.5.1. Sample Collection

On PND 33 or 34, pigs were euthanized via CO_2_ asphyxiation followed by exsanguination. Samples collected and stored for subsequent analyses included: serum, plasma, and tissue samples from the liver, ileum, ascending colon, and brain (pons, cerebellum, hippocampus, olfactory bulb, and prefrontal cortex dissected exclusively from the right hemisphere); aliquots of each tissue type were snap-frozen in liquid nitrogen and stored at −80 °C. Ascending colon lumenal contents and freshly-voided fecal samples were collected for volatile fatty acid (VFA) quantification as a measure of fermentative characteristics of the diet and an indirect indicator of microbial profiles.

#### 2.5.2. Volatile Fatty Acid (VFA) Analysis

Concentrations of VFA in ascending colon contents and feces were measured to understand possible gut-brain-axis mechanisms. Ascending colon content and freshly voided fecal samples were weighed and preserved using acidification with 6.25% m-Phosphoric acid. In brief, short chain fatty acid (SCFA) concentrations were determined using a gas chromatographer (Hewlett-Packard 5890A Series II) and a glass column (180 cm × 4 mm i.d.), packed with 10% SP-1200/1% H3PO4 on 80/100 + mesh Chromosorb WAW (Supelco/Sigma-Aldrich Corp., St. Louis, MO, USA). Nitrogen was used as the carrier gas with a flow rate of 48–52 mL/min. Oven, detector, and injector temperatures were set at 125 °C, 180 °C, and 175 °C, respectively.

### 2.6. Statistical Analysis

All data generated as part of this study were subjected to an analysis of variance (ANOVA) using the MIXED procedure of SAS 9.3 (SAS Inst. Inc., Cary, NC, USA). All statistical models included diet, litter, pig, and cohort as classification variables with cohort serving as the sole random variable. In every case, the individual pig served as the experimental unit, with a total of 14 pigs initially enrolled in the study to receive each of two dietary treatments. The level of significance was set at *p* < 0.05 for all comparisons. Depending on the outcome, one of two statistical models were used: (1) any data collected at a single time point (e.g., fecal samples) were analyzed by a simple ANOVA, and (2) any data collected from the same animal on more than one occasion (i.e., longitudinal parameter) were analyzed as a repeated-measures ANOVA. Parameters subjected to a repeated-measures ANOVA only included growth performance (e.g., daily body weights and milk intake). For some behavioral outcomes (e.g., comparing the NOR recognition index to a null value of 0.50), a one-sample t-test was conducted. All other parameters were subjected to a simple one-way ANOVA to statistically differentiate the effects of control and test diets that were provided to young pigs.

In terms of MRI outcomes, voxel-based morphometry analyses were performed using a two-sample t-test to compare control vs. test diets. An analysis of covariance (ANCOVA) was used for global normalization. Statistical maps using pseudo-t values were generated to show regional differences in gray and white matter between the two treatment groups. An uncorrected alpha of 0.01 was used to ensure proper statistical significance for individual comparisons. An additional threshold of at least 20 edge-connected voxels was used to count a voxel cluster as significant.

## 3. Results

### 3.1. Growth Performance and Health

In total, twenty-eight piglets were initially enrolled on study and assigned to two dietary treatment groups. Seven pigs were excluded from collecting MRI and behavioral outcomes due to failure-to-thrive (i.e., insufficient weight gain), so data from twenty-one pigs were used for subsequent analysis (*n* = 11 pigs in Control group and *n* = 10 pigs in OPN group). A main effect of age (*p* < 0.001), but no main effect of diet (*p* > 0.05), was observed for daily body weights and milk intake as shown in Figure 1. There were no effects of diet for average daily gain (ADG), average daily milk intake (ADMI), or gain-to-feed ratio (G:F; i.e., feed efficiency) at any interval assessed (all *p* > 0.63; Table 1).

### 3.2. VFA Concentration

Concentrations of VFA were measured to evaluate possible gut-brain-axis mechanisms. No dietary differences for concentrations of VFA in the ascending colon lumenal contents or feces were observed. However, there was a diet effect for the relative proportions of propionate:total SCFA (*p* = 0.018), valerate:total branched chain fatty acid (BCFA) (*p* = 0.046), and isobutyrate:total BCFA (*p* = 0.037) in ascending colon lumenal content samples (Table 2). No dietary effects on VFA concentrations were observed for fecal samples.

### 3.3. Novel Object Recognition

Behavioral outcomes from the NOR task were measured to assess the effects of OPN on cognitive development. First, a one-sample t-test comparing the recognition index with a null performance level of 0.50 was conducted to determine whether Control or Test group pigs were able to demonstrate novelty preference. Pigs fed the control diet demonstrated a novelty preference (*p* < 0.05) (i.e., they exhibited recognition memory) whereas pigs on the test diet did not (*p* > 0.05; Figure 2). A separate one-way ANOVA was conducted to compare the two diet groups to examine the effects of OPN ingestion. No dietary effects were observed for the recognition index (*p* > 0.05), as is evident in Figure 3. In other words, the capacity to learn was equivalent in Control and Test pigs, even though Control pigs performed above chance level in the NOR task. However, pigs receiving bovine milk OPN exhibited lower (*p* < 0.05) latency to the first exploration of an object compared with pigs on the control diet. No other exploratory behavior measurements were differed between dietary treatment groups (Table 3).

### 3.4. Brain Volume

The absolute volume of twenty-seven brain regions were measured to evaluate the influence of OPN on regional brain growth. There was no effect of diet on the absolute volume of the whole brain, gray matter, white matter, cerebrospinal fluid, nor any individual brain region (Appendix A). Pigs provided the test diet had larger relative volumes (i.e., % of total brain volume) of the corpus callosum (*p* = 0.013), left internal capsule (*p* = 0.017), right internal capsule (*p* = 0.002), left putamen-globus pallidus (*p* = 0.015), right putamen-globus pallidus (*p* = 0.012), right cortex (*p* = 0.020), right hippocampus (*p* = 0.024), and the lateral ventricle (*p* = 0.026) compared with control pigs (Table 4).

### 3.5. Diffusion Tensor Imaging

White matter integrity and overall brain microstructure were investigated utilizing DTI. No significant differences due to dietary treatment were observed for axial diffusivity (Appendix A) or mean diffusivity (Appendix A) outcomes. Radial diffusivity (Appendix A) was higher (*p* = 0.008) in the corpus callosum of pigs provided the test diet, which likely contributed to the fractional anisotropy (Table 5) value of this region being lower (*p* = 0.020) in pigs fed the test diet.

### 3.6. Voxel-Based Morphometry

For further evaluation of microstructural differences in brain anatomy and potential neuroanatomical differences between hemispheres, VBM was performed. Analysis of gray and white matter tissue segmentations revealed minor differences in regional tissue concentrations between treatment groups (Table 6). A comparison of gray matter concentration where control pigs had higher (*p* < 0.05) regional peak intensities of gray matter compared with test pigs (Control > Test) revealed differences in the right cortex, which had a cluster of 55 voxels. When analyzing regional clusters in which gray matter in test pigs was more concentrated than in control pigs (Test > Control), significant peak intensities in the right cortex and brainstem were evident, but had a small number of significant voxels (27 and 23 voxels, respectively). White matter concentration in which Control > Test resulted in 21 voxels in a cluster within the brainstem. No voxel cluster differences in white matter concentration were observed when Test > Control comparisons were analyzed.

## 4. Discussion

Human milk, especially early milk, is known to contain bioactive whey proteins, including OPN, at relatively high concentrations [6,10]. Various studies have shown that dietary bovine milk OPN supplemented in infant formulas has numerous beneficial effects on the development of immune function and intestinal health [15,40]. However, the ability for early-life bovine milk OPN to influence brain and cognitive development has not been extensively studied until recently [18,23]. Thus, the overall objective of the current study was to investigate the effects of a dietary bovine milk OPN-enriched supplement on brain and cognitive development during early life, especially during infancy, using the young pig as a translational model. Milk intake and growth performance outcomes were measured to evaluate safety of the dietary bovine milk OPN supplementation during early-life, and VFA were used to assess possible gut–brain–axis mechanisms. The influence of OPN ingestion on brain development was examined using structural and microstructural neuroimaging procedures and the NOR task was used to assess the effects of OPN on cognitive development. It is important to note that cognitive outcomes cannot always be inferred from neuroimaging measures, and vice versa, especially when there are minimal significant results observed (i.e., structure and function are not exclusively linked). Our results indicate that bovine milk OPN did not negatively impact growth or health outcomes but did influence colonic fermentation outcomes including the molar ratios of propionate and valerate. The effects of OPN on cognitive assessments were minimal, as the learning demonstrated in the Control group pigs was not significantly different from the Test group pigs. However, ingestion of the OPN-enriched ingredient increased relative volumes of several brain regions and elicited changes in exploratory behavior of young pigs, thereby providing supportive evidence of this serving as a beneficial ingredient.

### 4.1. Milk Intake and Growth Performance

Growth performance did not differ between the control and test diet groups throughout the study. Lack of the dietary effect in body weight measurements between two groups suggests that OPN supplementation was well-tolerated when ingested continually for the first 4 weeks of life. Similar growth outcomes have been reported in a randomized, double-blinded clinical study with 240 infants in which formula containing supplemental OPN caused no differences in formula intake or overall growth and health of infants [6]. A study on rats also revealed no toxic effects of dietary OPN supplementation [41]. Combined, these findings suggest that supplementation of naturally occurring, bovine milk-derived OPN has no negative effects on growth, milk intake, or other general indicators of health and well-being.

### 4.2. VFA

No differences were observed in VFA concentration from either the ascending colon or feces of pigs receiving bovine milk OPN compared with control pigs. However, changes in the relative proportion of propionate, valerate, and isobutyrate in ascending colon lumenal contents were evident, suggesting minor shifts in microbiota profiles had likely occurred. In enterotoxigenic *E. coli*-infected weanling pigs, higher concentrations of acetate, propionate, butyrate, and valerate in ascending colon lumenal contents were observed in OPN-supplemented pigs [42]. In our study, we observed the opposite, with a reduced relative proportion of propionate in ascending colon lumenal contents. This may be interpreted as a consequence of slightly increased acetate concentration, which in turn would increase total SCFA and affect the proportion of propionate to total SCFA. We also observed increased valerate proportion to total BCFA, most likely due to slightly decreased total BCFA concentration, especially isobutyrate. Collectively, these data suggest that bovine milk OPN ingested at concentrations comparable to those found in human milk may remain within the lumen of the gastrointestinal tract and be fermented by bacteria within the colon. Our study did not quantify changes in microbial profiles in the pig, so further research is warranted to elucidate the role of the microbiota when bovine milk OPN is consumed.

Recent evidence suggests VFA likely plays a role in the gut–brain–axis and modulating various aspects of behavior, including exploratory behaviors [43]. A previous study from our lab on effects of dietary prebiotics (i.e., fermentable carbohydrates) suggested that there may be a correlation between ascending colon VFA concentrations and exploratory behaviors with an unknown mediator playing a role [44]. A negative correlation between ascending colon isobutyrate concentration and recognition memory was found, although it seems to be minimal compared to many aspects of exploratory measures [44]. Although we observed a decreased proportion of isobutyrate in the current study, the test group pigs did not exhibit improved recognition memory. It is possible that we were not able to observe improved recognition memory in OPN-supplemented pigs because this is a weak association. However, although latency to first exploration of any object was not specifically discussed in the previous study, many correlations between ascending colon metabolites and exploratory measures were observed [44]. Overall, altered VFA concentrations in ascending colon lumenal contents and latency to the first object visit from the current study may provide additional supporting evidence to the hypothesis that microbial metabolites may influence behavior indirectly.

### 4.3. Magnetic Resonance Imaging

Analysis of absolute whole brain and individual brain regional volumes reveled no differences due to diet, confirming that early-life bovine milk OPN did not influence absolute brain volumes in our model. However, pigs on the test diet exhibited larger relative volumes of the corpus callosum, both internal capsule and putamen-globus pallidus, the lateral ventricle, and the right cortex and hippocampus. Relative shifts in volumetric expansion of the brain during early life are the most appropriate and sensitive indicator of brain growth, as they consider the allometric relationship between the brain and body size. Given that one of the known functions of brain OPN in the rodent includes the promotion of cellular proliferation and differentiation [9], the larger relative volumes of individual regions suggest that OPN may cause greater synaptogenesis of certain regions. During early development, the brain overproduces neurons and synaptic connections, some of which are kept and strengthened while others are removed or pruned over time [45]. In humans, cortical gray matter volume increases during the pre-adolescent period followed by a decline post-adolescence [46]. Similarly, pigs reach a maximum growth rate of total brain volume at around four weeks of age, whereas other areas, such as the hippocampus and cerebellum, have a maximum growth rate at later ages [47]. As such, the timing of neuroimaging events is critical, and it remains possible that imaging pigs at the single time-point used in the current study did not coincide with the expected neurodevelopmental events surrounding pruning. Had the current study extended to a later time-point, decreases in relative brain volume sizes may have been observed and, therefore, corroborated the role of synaptogenesis in observed structural changes to the brain.

Diffusion tensor imaging outcomes confirmed that the microstructural organization of the corpus callosum was significantly different between the two diets. The corpus callosum is a broad bundle of fibers that joins the two hemispheres of the brain. It functions to transmit information between the hemispheres to perform complex and uniform tasks such as the integration of sensory and motor functions [48,49], memory [50], and vision [51]. The DTI measurements indicate higher RD and lower FA in the corpus callosum of pigs receiving bovine milk OPN compared with control pigs. Developmental studies utilizing DTI typically attribute an increase in RD, and concomitant decrease in FA, to a less-mature and less-myelinated brain [52]. This phenomenon is thought to occur because myelination creates a physical barrier to the movement of molecules across the neuronal membrane, thereby preventing the perpendicular diffusion of water and increasing more directional water movement (i.e., decreasing RD and increasing FA). However, DTI measurements only confirm that microstructural differences in tissue exist and cannot explain the precise biological reasoning behind those differences [53,54]. Wahl and colleagues found that in humans, fiber density is best represented through FA more so than myelination [55]. Moreover, the corpus callosum is sensitive to early-life nutritional interventions [56] possibly due to the late maturation and myelination of this structure [57]. Thus, the corpus callosum microstructural disparities between diets may be due to resources going towards synaptogenesis in the corpus callosum rather than contributing to myelination.

Voxel-based morphometry is a technique that characterizes subtle changes in brain structure that may be due to a variety of disease states [58,59], environmental factors [60,61], or interventions such as nutritional supplementation [34,62]. This method detects changes in the shape of clusters of voxels within the brain by normalizing images to the same stereotaxic space and subsequently performing a voxel-by-voxel comparison between treatment groups to detect tissue volume differences. VBM results from the current study indicate little deviation in gray and white matter concentrations between pigs assigned to each dietary treatment group. Most clusters barely passed the 20-voxel threshold that was applied, which was far smaller than the cluster sizes of 100–1000 that have been previously observed in VBM assessments of young pigs [34]. Thus, we conclude that the collective MRI outcomes that were applied to pigs align with previous research suggesting that OPN supplementation increased brain regional volume. However, the VBM results suggest that changes in gray and white matter concentrations appear to be widespread within the brain, instead of there being focal points where tissue concentrations were altered.

### 4.4. Behavior

In the present study, the role of bovine milk OPN on cognitive development was examined using a novelty preference behavioral task designed for the pig. There were no major differences observed in outcomes associated with recognition memory between the control and test diet groups, although only pigs receiving the control diet exhibited novelty preference with a recognition index that was different from 0.50.

The capacity for milk-derived OPN to influence early-life development of cognitive capacity has not been studied extensively. In recent studies, the passive avoidance test was performed to access cognitive development [18,23]. Better memory was observed in wild type mice receiving milk OPN compared with OPN knock-out mice that did not receive milk OPN using the passive avoidance task [18]. Similarly, mice that received dietary OPN exhibited better memory compared to mice that did not receive milk OPN [23]. However, the effect of milk OPN on memory was measured at PND 30 in mice in both studies, which resembles toddlerhood, rather than infancy, in the human context. In our experimental design, the NOR task was conducted at PND 30, which is a time-point that approximates infancy in humans, as 1 month in infants approximately equates to 1 week in young pigs with regard to the rate of absolute brain volume expansion [63]. Collectively, we conclude the effect of dietary bovine milk OPN supplement on cognitive development was minimal.

However, examination of general exploratory measures revealed shorter latency to the first object visit in test pigs. General exploratory behavior measurements often include the total and average amount of time exploring objects, but not necessarily the latency to the first object explored [64]. Renner and Rosenzweig categorized latency measures as “emotionality”, separately from “exploratory”, but did not further describe a specific kind of emotion [64]. Considering the latency measures may account for emotionality involving exploratory behavior, one explanation of the shorter latency observed in the current study is reduced anxiety-like behavior in pigs receiving bovine milk OPN.

Behavioral paradigms assessing anxiety, such as open-field tasks, are rather controversial for pigs because there is a lack of consensus on what behavioral measurements are the most suitable to accurately account for anxiety [65]. Generally, the number of visits or the time spent in the center compared with these measures in the periphery are observed since pigs are known to exhibit wall-hugging behavior, a tendency to spend more time in the periphery of the novel environment than in the open area at the center [66,67,68]. It is important to note that both familiar and novel objects were located at the center of the arena during the NOR task in the current study. The central location of objects may induce anxiety towards the open area in pigs, although it is unclear whether anxiety towards the center of arena remains after repeated exposure to the testing environment.

While number of visits and time spent in the center are often measured, temporal aspects of these exploratory events are not typically accounted for in the published literature. A study validating behavioral measures in open-field tasks measured object-directed exploration measures, including latency to approach both the feeder and a novel object in the testing environment, and pigs given azaperone, an anti-stress drug for pigs, exhibited reduced latency to the novel object [66]. Although the open-field task with a novel object is not identical to the NOR in the current study, this finding suggests that reduced latency to the first object visit may provide an insight to reduced anxiety-related exploratory behaviors. Nevertheless, the motivation or reasoning behind such behavior is rather challenging to interpret because other exploratory measures such as latency to the last object visit, duration, and frequency of the object investigation did not show any diet effects.

In addition, there is a lack of knowledge on the OPN concentration in porcine milk, and the concentration and timing of dietary OPN that pigs would naturally consume from sow’s milk is unclear. The current study chose the concentration of bovine milk OPN to closely mimic that found in human breastmilk. However, for the reasons described above, it is challenging to determine if the concentration of bovine milk OPN consumed by pigs in the current study was sufficient to elicit improvements in cognitive development.

## 5. Conclusions

Early-life brain and cognitive development were examined in pigs receiving a dietary supplementation of bovine milk-derived OPN. Dietary inclusion of bovine milk OPN did not negatively influence overall growth performance or health indices, indicating it was safe when fed continuously in a pig model. In terms of colonic fermentation outcomes, the relative proportions of microbial metabolites, including propionate, valerate, and isobutyrate, were influenced by OPN ingestion. The addition of dietary bovine milk OPN also increased the relative volumes of various brain regions that exhibit relatively high rates of postnatal growth, suggesting possible changes to synaptogenesis, but underlying mechanisms still need to be investigated. Changes in an exploratory behavior were observed in pigs provided bovine milk OPN, but changes due to diet were minimal.

## Figures and Tables

**Figure 1 nutrients-12-02206-f001:**
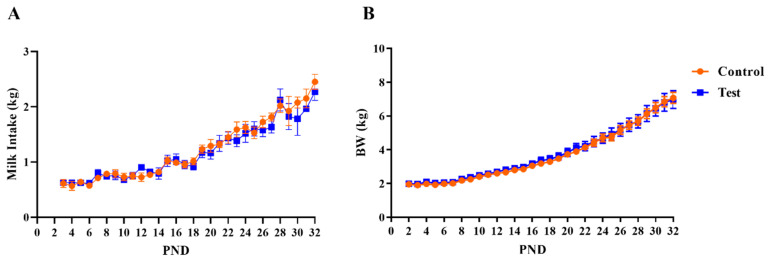
Growth performance over the study period, including (**A**) milk intake pattern and (**B**) body weight for pigs receiving control or test milk replacer formulations. PND 33 and 34 are not represented for milk intake due to fasting prior to euthanasia. Likewise, body weights for PND 34 are not represented as piglets were food-deprived the night prior. Abbreviations: BW, body weight; PND, postnatal day.

**Figure 2 nutrients-12-02206-f002:**
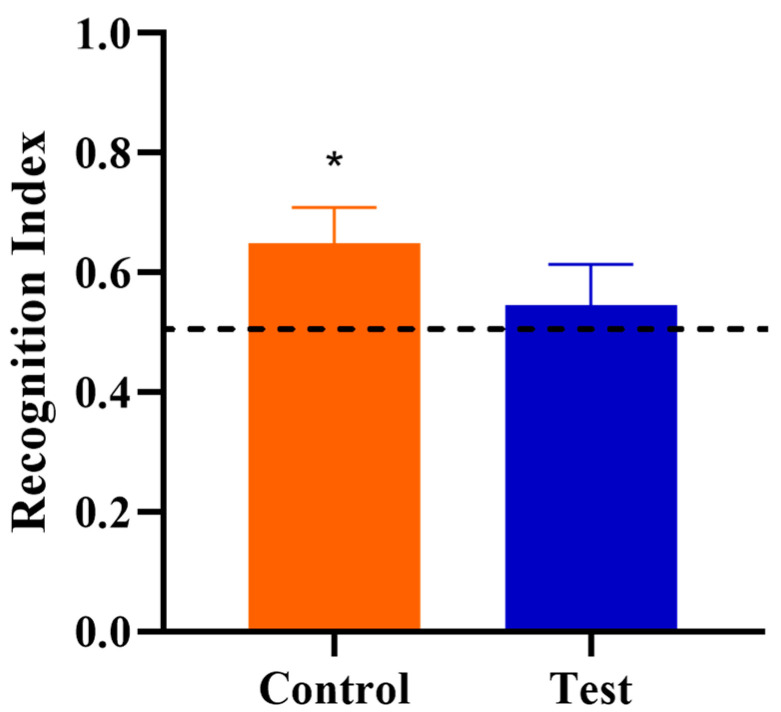
Recognition index during the test trial of the novel object recognition task as a measure of recognition memory. The dotted line at 0.50 indicates the chance performance value, and the recognition index greater than 0.50 indicates a novelty preference, and thus recognition memory. Note that an asterisk (*****) on the Control group denotes recognition memory as the recognition index was greater (*p* = 0.017) than 0.50. The Test group did not exhibit recognition memory (*p* = 0.264).

**Figure 3 nutrients-12-02206-f003:**
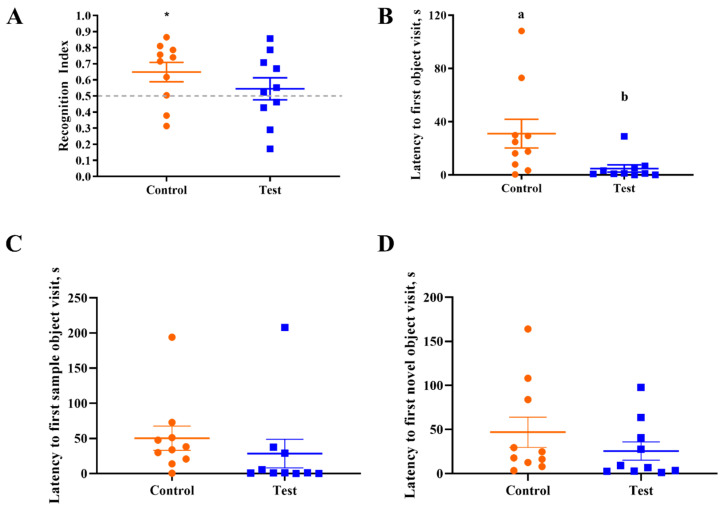
Performance on the test trial of the novel object recognition task. (**A**) Only the control group was able to demonstrate a novelty preference, no dietary effect between two treatment groups was observed. (**B**) Latency to the first object visit was significantly great in the control group. Note that an asterisk (*****) denotes a difference (*p* < 0.05) from the null RI value of 0.5 and means lacking a common superscript letter differ (*p* < 0.05). Thus, superscript letters (^a b^) denote differences between treatment means. (**C**,**D**) Latency to the first sample and novel object visit were not different between the control and test groups.

**Table 1 nutrients-12-02206-t001:** Growth and feeding performance PND 3–33 ^1^.

	Diet		Pooled SEM	*p*-Value ^2^
Measure	Control	Test		
ADG, g/day	195	188	22.4	0.70
ADMI, g liquid milk/day	1225	1183	111.8	0.63
G:F, g BWG:g liquid milk intake	0.159	0.156	0.005	0.71

^1^ Pigs received diets containing 0 (Control) or 250 (Test) mg of bovine milk OPN per L of mixed milk replacer during a 30d feeding study (*n* = 11 male pigs for control group and 10 male pigs for test group). Abbreviations: ADG, average daily body weight gain; ADMI, average daily milk intake; BWG, body weight gain; G:F, feed efficiency; OPN, osteopontin; PND, postnatal day; SEM, standard error of the mean. ^2^
*p*-values derived from mixed model ANOVA.

**Table 2 nutrients-12-02206-t002:** Concentration of volatile fatty acids (VFA) in ascending colon contents and feces ^1^.

	Diet		Pooled SEM	*p*-Value
Item	Control	Test		
Ascending Colon				
DM, %	23.77	23.95	0.677	0.849
SCFA absolute, μmol/g DM				
Acetate	47.34	56.40	6.043	0.304
Propionate	23.72	23.05	2.831	0.869
Butyrate	9.82	10.35	1.442	0.798
Total SCFA	80.88	89.80	9.800	0.528
SCFA relative, % of total				
Acetate	59.12	63.00	2.334	0.067
Propionate	28.92	25.75	1.660	**0.018**
Butyrate	11.95	11.27	0.747	0.479
BCFA absolute, μmol/g DM				
Isovalerate	5.61	4.32	0.703	0.203
Valerate	2.48	2.47	0.298	0.976
Isobutyrate	3.76	2.80	0.444	0.136
Total BCFA	12.30	9.58	1.553	0.222
BCFA relative, % of total				
Isovalerate	45.96	44.96	1.808	0.319
Valerate	23.12	26.18	2.547	**0.046**
Isobutyrate	30.90	28.86	0.846	**0.037**
Feces				
DM, %	30.15	32.00	1.922	0.437
SCFA absolute, μmol/g DM				
Acetate	43.09	30.33	4.953	0.080
Propionate	15.59	9.80	2.912	0.111
Butyrate	5.91	3.83	1.586	0.154
Total SCFA	73.39	43.95	11.30	0.051
SCFA relative, % of total				
Acetate	68.82	70.57	3.516	0.606
Propionate	21.40	21.43	1.362	0.986
Butyrate	9.86	7.99	2.309	0.294
BCFA absolute, μmol/g DM				
Isovalerate	5.07	3.67	0.825	0.248
Valerate	2.31	1.48	0.466	0.116
Isobutyrate	3.20	2.33	0.485	0.224
Total BCFA	10.61	7.49	1.671	0.195
BCFA relative, % of total				
Isovalerate	48.00	48.81	0.927	0.545
Valerate	21.56	19.79	1.349	0.305
Isobutyrate	30.45	31.40	0.815	0.279

^1^ Pigs received diets containing 0 (Control) or 250 (Test) mg of bovine milk OPN per L of mixed milk replacer during a 30-d feeding study (*n* = 11 male pigs for control group and 10 male pigs for test group). Bolded *p*-values denote differences between treatment means (*p* < 0.05). Abbreviations: BCFA, branched-chain fatty acids; DM, dry matter; OPN, osteopontin; SCFA, short-chain fatty acids; SEM, standard error of the mean; VFA, volatile fatty acids.

**Table 3 nutrients-12-02206-t003:** Exploratory behavior during the test trial of the novel object recognition (NOR) task ^1^.

	Control		Test		Pooled	
Measures During the Test Trial	*n*	Mean	*n*	Mean	SEM	*p*-Value ^2^
Recognition Index	10	0.65	10	0.55	0.064	0.269
Exploration of the novel object						
Novel object visit time, s	11	36.0	10	43.8	11.19	0.617
Number of novel object visits	11	8.3	10	9.6	1.72	0.584
Mean novel object visit time, s	10	4.1	10	4.3	0.86	0.839
Latency to first novel object visit, s	10	46.8	10	25.5	14.08	0.299
Latency to last object visit, s	10	220.5	10	260.1	13.90	0.059
Standard error of novel object visit time, s/visit	9	1.5	10	1.5	0.35	0.981
Exploration of the sample object						
Sample object visit time, s	11	17.0	10	44.3	10.56	0.077
Number of sample object visits	11	6.1	10	7.4	1.34	0.487
Mean sample object visit time, s/visit	10	2.6	10	5.6	1.20	0.093
Latency to first sample object visit, s	10	50.3	10	28.4	18.88	0.424
Latency to last sample object visit, s	10	212.4	10	259.7	20.86	0.126
Standard error of sample object visit time, s/visit	9	1.0	10	1.9	0.41	0.120
Exploration of all objects						
Total object visit time, s	11	53.0	10	88.1	17.80	0.170
Number of object visits	11	14.4	10	17.0	2.57	0.466
Mean object visit time, s/visit	10	3.4	10	4.9	0.830	0.234
Latency to first object visit, s	10	31.1	10	4.9	7.84	**0.029**
Latency to last object visit, s	10	249.9	10	277.3	11.45	0.108
Standard error of total visit time, s/visit	9	1.5	10	1.5	0.35	0.981

^1^ Pigs received diets containing 0 (Control) or 250 (Test) mg of bovine milk OPN per L of mixed milk replacer during a 30-d feeding study (*n* = 11 male pigs for control group and 10 male pigs for test group). Bolded *p*-values denote differences between treatment means (*p* < 0.05). Abbreviation: OPN, osteopontin; SEM, standard error of mean. ^2^
*p*-value derived from F-test for the fixed effect of dietary treatment.

**Table 4 nutrients-12-02206-t004:** Relative brain volumes (% total brain volume) ^1^.

**Region of Interest**	**Diet**		**Pooled SEM**	***p*** **-Value**
	**Control**	**Test**		
Gray matter	66.79	66.66	3.663	0.974
White matter	25.49	26.27	1.104	0.616
Cerebrospinal fluid	7.34	7.06	3.574	0.931
Cerebellum	10.16	10.09	0.561	0.827
Cerebral aqueduct	0.02	0.02	0.001	0.628
Corpus callosum	0.39	0.41	0.005	**0.013**
Fourth ventricle	0.03	0.03	0.001	0.672
Hypothalamus	0.15	0.15	0.003	0.519
Lateral ventricle	0.54	0.57	0.009	**0.026**
Left caudate	0.35	0.36	0.007	0.105
Left cortex	26.08	27.10	0.428	0.093
Left hippocampus	0.47	0.48	0.007	0.115
Left inferior colliculi	0.11	0.11	0.004	0.932
Left internal capsule	0.81	0.86	0.018	**0.017**
Left olfactory bulb	1.87	1.98	0.060	0.201
Left putamen-globus pallidus	0.19	0.20	0.007	**0.015**
Left superior colliculi	0.27	0.27	0.005	0.869
Medulla	2.42	2.45	0.121	0.715
Midbrain	3.39	3.39	0.069	0.974
Pons	2.04	2.04	0.079	0.971
Right caudate	0.36	0.37	0.005	0.098
Right cortex	25.93	26.91	0.278	**0.020**
Right hippocampus	0.49	0.52	0.009	**0.024**
Right inferior colliculi	0.11	0.11	0.003	0.943
Right internal capsule	0.78	0.84	0.011	**0.002**
Right olfactory bulb	1.84	1.95	0.056	0.171
Right putamen-globus pallidus	0.18	0.19	0.003	**0.012**
Right superior colliculi	0.28	0.29	0.004	0.614
Thalamus	1.82	1.87	0.022	0.156
Third ventricle	0.03	0.03	0.001	0.665

^1^ Pigs received diets containing 0 (Control) or 250 (Test) mg of bovine milk OPN per L of mixed milk replacer during a 30-d feeding study (*n* = 11 male pigs for control group and 10 male pigs for test group). Bolded *p*-values denote differences between treatment means (*p* < 0.05). Abbreviations: OPN, osteopontin; SEM, standard error of mean.

**Table 5 nutrients-12-02206-t005:** Fractional Anisotropy (FA; arbitrary units) ^1^.

Region of Interest	Diet		Pooled SEM	*p*-Value
	Control	Test		
Corpus callosum	0.36	0.31	0.036	**0.020**
Cerebellum	0.23	0.24	0.005	0.359
Left caudate	0.31	0.31	0.013	0.602
Left hippocampus	0.35	0.34	0.014	0.655
Left internal capsule	0.56	0.55	0.021	0.882
Right caudate	0.29	0.30	0.011	0.813
Right hippocampus	0.35	0.35	0.011	0.571
Right internal capsule	0.56	0.55	0.028	0.833
Left side	0.36	0.36	0.002	0.599
Right side	0.36	0.36	0.002	0.714
Thalamus	0.29	0.30	0.005	0.451
T1 white matter	0.37	0.37	0.003	0.744
Average FA mask	0.32	0.33	0.002	0.850

^1^ Pigs received diets containing 0 (Control) or 250 (Test) mg of bovine milk OPN per L of mixed milk replacer during a 30-d feeding study (*n* = 11 male pigs for control group and 10 male pigs for test group). Bolded *p*-value denote differences between treatment means (*p* < 0.05). Data presented are least square means and P-values from mixed model ANOVA. Abbreviation: OPN, osteopontin; SEM, standard error of mean.

**Table 6 nutrients-12-02206-t006:** Tissue density differences within clusters from -based morphometry assessment ^1^.

Tissue	Comparison ^2^	Anatomic Region ^3^	Cluster, Number of Voxels ^4^	Peak-Level		Local Maxima Coordinates ^5^		
				Pseudo-t	*p*-Value	X	Y	Z
Gray	Control > Test	Right Cortex	55	3.21	0.0077	16	3	14
Gray	Test > Control	Right Cortex	27	1.70	0.0087	22	−3	3
Gray	Test > Control	Brainstem	23	0.49	0.0088	8	−9	−15
White	Control > Test	Brainstem	21	1.48	0.0084	6	−8	−15

^1^ Pigs received diets containing 0 (Control) or 250 (Test) mg of bovine milk OPN per L of mixed milk replacer during a 30d feeding study (*n* = 11 male pigs for control group and 10 male pigs for test group). Voxel-based morphometry analysis of gray and white matter differences in the Test and Control piglet brains. A threshold of *p* < 0.01 and minimum cluster size of 20 voxels were used to determine *p*-uncorrected values listed in the table. Abbreviation: OPN, osteopontin.^2^ Indicates which diet had greater tissue density for the corresponding cluster defining a set area within the standardized atlas space. ^3^ Brain regions based on estimates from the University of Illinois Piglet Brain Atlas. ^4^ Clusters are defined as 3-dimentional volumes composed of contiguous voxels within a static location within the standardized atlas space. ^5^ Local maxima coordinates: x increases from left (−) to right (+), y increases from posterior (−) to anterior (+), and z increases from inferior (−) to superior (+).

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
