# Peer review of "Early-Life Supplementation of Bovine Milk Osteopontin Supports Neurodevelopment and Influences Exploratory Behavior"

_nutrients, 2020, doi:10.3390/nu12082206_

Round 1

Reviewer 1 Report

The authors investigated effects of oral intake of bovine milk osteopontin on cognitive development in early life using the piglet as a translational model by conducting MRI analysis and a novel subject recognition task.

A major criticism relates to why the piglet model was chosen for the current study. Are there any advantages of the piglet model compared with the other animal models? Does sow milk contain OPN? Does sow milk contain a high concentration of OPN? Is bovine OPN similar to sow OPN? Glycosylations? Phosphorylations? This model needs to be validated prior to evaluating potential bioactivities of orally provided bovine osteopontin.

Specific criticisms:

  1. In the Abstract, the authors claim that “the influence of dietary bovine milk inclusion on brain and cognitive development has not been studied in depth”. This is not correct, and they later cite references illustrating this. In fact, those studies are more in depth than their own study, as it does not explore any mechanisms and is observational (with fairly modest effects found).
  2. Lines 60-62. These references do not show that early human milk is especially high in OPN. Ref 7 is a review article and ref. 8 shows OPN in a limited number of mature milk samples. Ref 9 consists of microarray results showing high expression throughout lactation.
  3. Lines 84-86. “Infant monkeys did not exhibit differences….” What monkeys? Which groups” Clarify!
  4. Lines 119-120, what are 2 replicates? Were piglets from the same litter allotted to both control and OPN groups? Were 21 or 28 piglets used in the present study? If 21 piglets were included in the current study, each group did not have equal number of piglets from the same litter.
  5. How long does it take to wean a pig? Why was the timeline (PND 2-34) selected for the current study?
  6. Lines 161-162,testing on the NOR task began at PND 30?
  7. Figure 2, can this result be interpreted as the control group showing better recognition memory than the test group?
  8. Were Figure 2 and Figure 3A generated from the same set of results? What is the difference between Figure 2 and Figure 3A? Line 314, Figure 3B legend, what is the first object visit? Does the object include both sample object and novel object?
  9. The authors assume that increased relative brain volumes are positively related to better cognitive development. Are there any examples from the literature showing that increased relative brain volumes are positively associated with better cognitive development? How does the enlarged relative brain volume reflect the altered behaviors in the NOR task?
  10. What are the possible mechanisms underlying the above observations?
  11. Line 370. Please avoid the term “human breastmilk” – it is redundant. It is either human milk or breast milk (only humans have breast milk, there is no sow breastmilk”
  12. Table 6, although the authors claim that only group differences in clusters of voxels were indicated in the Material and Methods section, it is hard to see under the column (Cluster # voxels). For example, in the first row, does 55 mean that the control group has 55 more voxels in right cortex than does the test group?
  13. For all the tables, it would be clearer that results with significant differences are highlighted with either labels or other approaches although the p-value shows the difference. Some tables contain many items, and only a couple of items show remarkable differences. For example, Table 2 & 4.
  14. There are many grammatical errors throughout, so the English needs correction.
  15. The references seem to be hastily put together – there are several formats used (titles capitalized, titles not capitalized). Ref 11 seems to have one of the authors’ building cited, Ref 15 seems to have an odd title.

Author Response

Review #1

The authors investigated effects of oral intake of bovine milk osteopontin on cognitive development in early life using the piglet as a translational model by conducting MRI analysis and a novel subject recognition task.

A major criticism relates to why the piglet model was chosen for the current study. Are there any advantages of the piglet model compared with the other animal models? Does sow milk contain OPN? Does sow milk contain a high concentration of OPN? Is bovine OPN similar to sow OPN? Glycosylations? Phosphorylations? This model needs to be validated prior to evaluating potential bioactivities of orally provided bovine osteopontin.

A: We have included the advantages of using neonatal pig as a biomedical model for nutritional intervention researches in the introduction (line 108-117). Additionally, a lack of knowledge on the OPN concentration in porcine milk was discussion at the end of discussion section (line 515-520).

Specific criticisms:

  • In the Abstract, the authors claim that “the influence of dietary bovine milk inclusion on brain and cognitive development has not been studied in depth”. This is not correct, and they later cite references illustrating this. In fact, those studies are more in depth than their own study, as it does not explore any mechanisms and is observational (with fairly modest effects found).
    1. We have changed the verbiage so it remains consistent with statements made in the introduction section.

  • Lines 60-62. These references do not show that early human milk is especially high in OPN. Ref 7 is a review article and ref. 8 shows OPN in a limited number of mature milk samples. Ref 9 consists of microarray results showing high expression throughout lactation.
    1. We omitted the “early milk” phrasing as the main purpose of this sentence is to address that human milk contains higher concentrations of OPN compared with commercial infant formulas. The references have been adjusted accordingly.

  • Lines 84-86. “Infant monkeys did not exhibit differences….” What monkeys? Which groups” Clarify!
    1. Thank you for this suggestion. We have clarified that these are infant rhesus monkeys who were fed the infant formula supplemented with bovine milk OPN.

  • Lines 119-120, what are 2 replicates? Were piglets from the same litter allotted to both control and OPN groups? Were 21 or 28 piglets used in the present study? If 21 piglets were included in the current study, each group did not have equal number of piglets from the same litter.
  • How long does it take to wean a pig? Why was the timeline (PND 2-34) selected for the current study?
    1. In commercial settings, pigs are typically weaned at 18-24 days of age. In current study, pigs were technically weaned at PND 2, but we provided them with optimal rearing environment as described in ref 31 (1). Pigs remained on study through PND 34 because pigs experience a rapid brain development during 30 – 40 days after birth (2). Also, NOR in pigs has been validated previously by our lab, and it was suggested that the NOR task is suitable for pigs as young as 2 weeks of age (3).

  • Lines 161-162, testing on the NOR task began at PND 30?
    1. The NOR task we used in the current study is a previous validated behavioral task designed for pigs (3). The language referred to by the reviewer was consistent with our previous work using the NOR in pigs. However, we acknowledge how it can be confusing, so we omitted the sentence. Instead, we noted the postnatal days of age for each NOR phase on lines 161, 162 and 165.

  • Figure 2, can this result be interpreted as the control group showing better recognition memory than the test group?
    1. Figure 2 is comparing recognition index of control and test groups to 0.50, the chance performance value, to measure recognition memory. It is not a comparison between the two dietary treatment groups. What Figure 2 demonstrates is that Control pigs were able to exhibit novelty preference (i.e., they learned to distinguish the familiar object versus the novel object) while the Test pigs were unable to do so. Figure 3A is a comparison between the two diets and demonstrates no significant effect of diet on recognition index meaning that diet does not influence whether a pig is able to remember the familiar object versus the novel one.

Overall, there was no effect of dietary treatment on recognition index, even though the Control pigs demonstrated novelty preference and the Test pigs did not. Thus, we did not interpret this as the control group exhibiting better recognition memory or deficits in behavioral outcomes due to the OPN supplementation. Because of these findings, we are not comfortable over-interpreting our data to suggest that Test pigs were at a cognitive deficit relative to Control pigs.

  • Were Figure 2 and Figure 3A generated from the same set of results? What is the difference between Figure 2 and Figure 3A? Line 314, Figure 3B legend, what is the first object visit? Does the object include both sample object and novel object?
    1. Both Figure 2 and Figure 3A are generated from recognition index. However, Figure 2 is comparing recognition index to the chance performance level, 0.50. Figure 3A is a comparison between two treatment groups. See above comment for more detailed answer. The first object visit includes both sample and novel objects. Therefore, the latency to first object visit refers to the amount of time pigs took from the start of the trial to the first exploratory event to any objects.

  • The authors assume that increased relative brain volumes are positively related to better cognitive development. Are there any examples from the literature showing that increased relative brain volumes are positively associated with better cognitive development? How does the enlarged relative brain volume reflect the altered behaviors in the NOR task?
  • What are the possible mechanisms underlying the above observations?
    1. To answer these two questions simultaneously, we provided additional explanation at the beginning of the discussion section in order to appropriately set the tone for the rest of the discussion. Although we did observe increased relative brain volumes in certain brain regions, we were careful not to infer improved cognition (i.e., structure does not necessarily dictate function). Generally, the correlation between brain volume outcomes and cognitive behaviors is rather controversial with widely varying results and different hypotheses (4). Given minimal significant findings in neuroimaging and cognitive assessments, we are not comfortable making claims that directly related brain structure and cognition performance on the NOR behavioral task.

  • Line 370. Please avoid the term “human breastmilk” – it is redundant. It is either human milk or breast milk (only humans have breast milk, there is no sow breastmilk”
    1. We appreciate this comment and have edited this to read “human milk”.

  • Table 6, although the authors claim that only group differences in clusters of voxels were indicated in the Material and Methods section, it is hard to see under the column (Cluster # voxels). For example, in the first row, does 55 mean that the control group has 55 more voxels in right cortex than does the test group?
    1. Language was added to clarify the interpretation of the table. The number of voxels just expresses the size of the cluster (region) that has a greater tissue density. We have done our best to inform the reader that a cluster of 55 voxels means there is a defined area in standardized atlas space where pigs on the two treatment diets differed in grey matter concentrations. To clarify, the exact same 55-voxel space was assessed across all pigs on study, and it was within this defined cluster where concentration differences were noted.

  • For all the tables, it would be clearer that results with significant differences are highlighted with either labels or other approaches although the p-value shows the difference. Some tables contain many items, and only a couple of items show remarkable differences. For example, Table 2 & 4.
    1. Thank you for this suggestion, and significant P-values in all tables have now been bolded to emphasize significant differences.

  • There are many grammatical errors throughout, so the English needs correction.
    1. We appreciate the comment, and we have had an independent third party review the entire manuscript again for grammatical errors.

  • The references seem to be hastily put together – there are several formats used (titles capitalized, titles not capitalized). Ref 11 seems to have one of the authors’ building cited, Ref 15 seems to have an odd title.
    1. We carefully examined all references and corrected the ones that had inappropriate formats and information.

References

  1. Mudd AT, Alexander LS, Berding K, Waworuntu R V., Berg BM, Donovan SM, et al. Dietary Prebiotics, Milk Fat Globule Membrane, and Lactoferrin Affects Structural Neurodevelopment in the Young Piglet. Front Pediatr (2016) 4:1–10. doi:10.3389/fped.2016.00004
  2. Sweasey D, Patterson DS, Glancy EM. Biphasic myelination and the fatty acid composition of cerebrosides and cholesterol esters in the developing central nervous system of the domestic pig. J Neurochem (1976) 27:375–80. Available at: http://www.ncbi.nlm.nih.gov/pubmed/965978 [Accessed July 2, 2018]
  3. Fleming SA, Dilger RN. Young pigs exhibit differential exploratory behavior during novelty preference tasks in response to age, sex, and delay. Behav Brain Res (2017) 321:50–60. doi:10.1016/j.bbr.2016.12.027
  4. Van Petten C. Relationship between hippocampal volume and memory ability in healthy individuals across the lifespan: Review and meta-analysis. Neuropsychologia (2004) 42:1394–1413. doi:10.1016/j.neuropsychologia.2004.04.006

Reviewer 2 Report

The paper “Early-Life Supplementation of Bovine Milk Osteopontin Supports Neurodevelopment and Influences Exploratory Behavior” by Joung et al. claims to find a benefit on neurodevelopment by Osteopontin. However, the message conveyed is highly inconsistent. On the one hand, neuroimaging suggests increased relative brain volumes of the corpus callosum, lateral ventricle, internal capsules, putamen-globus pallidus, right hippocampus, and right cortex by Osteopontin. On the other hand, the dietary supplement seems to reduce novelty preference and the recognition index. Whereas the latency to first object visit is reduced by Osteopontin, the sample object visit time (although falling short of reaching statistical significance) seems prolonged.

The impact on microbial metabolites in the gut presumably was tested to assess safety, but that is not clearly stated.

The manuscript contains several grammatical errors, including incomplete and erroneous statements.

This research examines whether dietary supplementation of bovine milk OPN supports brain and cognitive development. It measures a variety of metabolic, behavioral, and brain morphological readouts. Seemingly, there is a discrepancy between purportedly favorable brain development morphology and deficits in behavioral tests. While it is not at all apparent how the diverse data sets can lead to meaningfully coherent conclusions, the authors make no effort to offer them to the reader. The paragraphs of the Results section are short. They would benefit from a rationale why certain experiments were done, as well as from a few words of explanation how they fit in with the prior analyses. In the discussion, the reference to the gut-brain-axis is too circumscript to meaningfully feed into conclusions. Most of the discussion is verbose without offering meaningful conclusions to draw.

Author Response

Review #2

  1. The paper “Early-Life Supplementation of Bovine Milk Osteopontin Supports Neurodevelopment and Influences Exploratory Behavior” by Joung et al. claims to find a benefit on neurodevelopment by Osteopontin. However, the message conveyed is highly inconsistent. On the one hand, neuroimaging suggests increased relative brain volumes of the corpus callosum, lateral ventricle, internal capsules, putamen-globus pallidus, right hippocampus, and right cortex by Osteopontin. On the other hand, the dietary supplement seems to reduce novelty preference and the recognition index. Whereas the latency to first object visit is reduced by Osteopontin, the sample object visit time (although falling short of reaching statistical significance) seems prolonged.
    1. For the behavioral outcomes, the dietary OPN-enriched ingredient did not necessarily reduce novelty preference, given that no significant diet effect was observed for recognition index. However, we acknowledge that may be confusing, so we provided additional explanation in NOR results section (line 300-310) to clarify.
    2. The shortened latency to the first object visit is an interesting finding that we addressed in the discussion section in relation to anxiety-related behaviors. However, we agree that it is challenging to relate our behavioral findings to the changes we see in brain morphology, and we added clarification of this point in the opening paragraph of the discussion section.

  1. The impact on microbial metabolites in the gut presumably was tested to assess safety, but that is not clearly stated.
    1. We thank the reviewer for this comment and we provided a rationale why we measured VFA on lines 244 and 297. The VFA analysis was conducted to assess possible gut-brain-axis mechanisms, as previous work from our lab has discussed these potential mechanisms (1). Growth and health outcomes were used to assess safety.

  1. The manuscript contains several grammatical errors, including incomplete and erroneous statements.
    1. We appreciate the comment, and we have had an independent third party review the entire manuscript for grammatical errors.

  1. This research examines whether dietary supplementation of bovine milk OPN supports brain and cognitive development. It measures a variety of metabolic, behavioral, and brain morphological readouts. Seemingly, there is a discrepancy between purportedly favorable brain development morphology and deficits in behavioral tests. While it is not at all apparent how the diverse data sets can lead to meaningfully coherent conclusions, the authors make no effort to offer them to the reader.
    1. We intentionally assessed a variety of readouts across domains and acknowledge that with diverse findings and minimal statistically significant results, it can be challenging to draw coherent conclusions. However, we do not interpret the results from NOR task as deficits in cognitive development. As described in the results section, we did not observe the diet effect in recognition index, which suggests that the learning from Control and Test groups are not significantly different. Furthermore, potential benefits of dietary OPN in reducing anxiety related behavior are discussed as well. Because of these findings, we are not comfortable over-interpreting our data to suggest that Test pigs were at a cognitive deficit relative to Control pigs. Also, we provided additional explanation at the beginning of the discussion section to clarify that it is challenging to conclude neuroimaging outcomes can be used to infer cognitive assessments.

  1. The paragraphs of the Results section are short. They would benefit from a rationale why certain experiments were done, as well as from a few words of explanation how they fit in with the prior analyses.
    1. It is unclear to us what the reviewer is referring to as ‘prior analyses.’ However, we provided a rationale why each outcome was measured at the beginning of each results sub-section and hope this is sufficient to assist readers.

  1. In the discussion, the reference to the gut-brain-axis is too circumscript to meaningfully feed into conclusions. Most of the discussion is verbose without offering meaningful conclusions to draw.
    1. The main purpose of providing VFA measures was to examine potential gut-brain-axis mechanisms. However, we agree that the VFA-related discussion and conclusions are rather vague especially with no significant findings. We thought it was still worth discussing because our previous work with different prebiotics has discussed interesting potential mechanism between luminal content and exploratory behaviors as discussed in VFA discussion section (1).

References

  1. Fleming SA, Monaikul S, Patsavas AJ, Waworuntu R V., Berg BM, Dilger RN. Dietary polydextrose and galactooligosaccharide increase exploratory behavior, improve recognition memory, and alter neurochemistry in the young pig. Nutr Neurosci (2019) 22:499–512. doi:10.1080/1028415X.2017.1415280

Round 2

Reviewer 1 Report

The major criticism of this manuscript was the validity of the piglet model.

The authors don't really respond to this - they give reasons why the piglet model is a good for human infants when it comes to gastrointestinal physiology, brain development, etc. I fully agree with that!

The validity question was related to the presence of OPN in piglets and in sow's milk. If this protein is absent, this would NOT be a good model to study potential bioactivities of OPN. The authors have previously worked on lactoferrin, a protein high in human milk. This protein is also abundant in sow's milk, making the piglet a good model to study bioactivities of lactoferrin. In contrast, the rat, a commonly used animal model, is lacking lactoferrin in its milk and is therefore NOT a good model to study bioactivities of lactoferrin.

It would be easy to check the presence of OPN in sow's milk, either by Westerns using antibodies against bovine or human OPN, or - at least running PAGE gels checking bands at the molecular weight for OPN.

Author Response

We thank the reviewer for clarifying what information they felt was missing, and we are now understanding that this sentiment was directed specifically at whether OPN would be expected to work in the pig. To that end, we have added additional references to the manuscript which highlight the following points:

  • OPN expression does occur in porcine cells: https://nyaspubs.onlinelibrary.wiley.com/doi/abs/10.1111/j.1749-6632.1995.tb44615.x
  • OPN in pigs is a crucial signal for embryo implantation and is present at the maternal-fetal interface: https://academic.oup.com/biolreprod/article/66/3/718/2723913
  • OPN expression in pigs is upregulated by exposure to infectious agents: https://pubmed.ncbi.nlm.nih.gov/17178162/
  • Preterm pigs fed an OPN-enriched ingredient nearly identical to that used in our study exhibited dietary OPN-induced immune and gut responses during an inflammatory challenge: https://academic.oup.com/biolreprod/article/66/3/718/2723913

Thus, we believe the addition of these references will bolster the narrative of what is currently know about OPN in the pig and thereby satisfy the reviewer’s concern. This information has been added to the Introduction section on lines 108-114 of the revised manuscript, along with the addition of these references to the bibliography.

Finally, we agree with the reviewer wholeheartedly that an analysis of porcine colostrum and milk to confirm the presence of OPN is warranted, but that effort was not within the scope of the current study. However, we are currently planning another OPN study in pigs, and our research plan includes this stated milk analysis.

Reviewer 2 Report

The authors have made an effort to address the issues raised in the initial review. The edits have substantially strengthened the paper. 

Author Response

We appreciate all the effort the reviewer has put into evaluating our manuscript!